**Data Availability Statement:** Data cannot be publicly shared because of identifying information. The name of the ethics committee that imposed

# Development of a multivariate prediction model of intensive care unit transfer or death: A French prospective cohort study of hospitalized COVID-19 patients

Yves Allenbach[1‡]*, David Saadoun[1‡]*, Georgina Maalouf [1º], Matheus Vieira[1º], Alexandra Hellio[1], Jacques Boddaert[2], Hélène Gros[3], Joe Elie Salem[4], Matthieu Resche Rigon[5], Cherifa Menyssa[5], Lucie Biard[5‡], Olivier Benveniste[1‡], Patrice Cacoub[1‡], on behalf of DIMICOVID[¶]

**1** Department of Internal Medicine and Clinical Immunology, Centre National de Références Maladies Autoimmunes et Systémiques Rares Maladies Autoinflammatoires Rares et des Myopathies Inflammatoires, AP-HP, Groupe Hospitalier Pitié-Salpêtrière, Sorbonne University, Paris, France, **2** Department of Geriatrics, Sorbonne Université, AP-HP, Groupe Hospitalier Pitié-Salpêtrière, Paris, France, **3** Robert-Ballanger Hospital, Aulnay-sous-Bois, France, **4** Department of Pharmacology and Clinical Investigation Center, Sorbonne Université, AP-HP, Groupe Hospitalier Pitié-Salpêtrière, CIC Pitié-Salpêtrière (CIC-1901), Paris, France, **5** Department of Biostatistics and Medical Information, AP-HP Saint-Louis University Hospital; ECSTRRA Team, CRESS UMR 1153, INSERM, University of Paris, Paris, France

º These authors contributed equally to this work.
‡ YA, DS, LB, OB and PC are co-senior authors on this work. LB, OB and PC share co authorship on this work.
¶ Membership of DIMICOVID is provided in the Acknowledgments.
* yves.allenbach@aphp.fr (YA); david.saadoun@aphp.fr (DS)

## Abstract

Prognostic factors of coronavirus disease 2019 (COVID-19) patients among European population are lacking. Our objective was to identify early prognostic factors upon admission to optimize the management of COVID-19 patients hospitalized in a medical ward. This French single-center prospective cohort study evaluated 152 patients with positive severe acute respiratory syndrome coronavirus 2 real-time reverse transcriptase–polymerase chain reaction assay, hospitalized in the Internal Medicine and Clinical Immunology Department, at Pitié-Salpêtrière's Hospital, in Paris, France, a tertiary care university hospital. Predictive factors of intensive care unit (ICU) transfer or death at day 14 (D14), of being discharge alive and severe status at D14 (remaining with ventilation, or death) were evaluated in multivariable logistic regression models; models' performances, including discrimination and calibration, were assessed (C-index, calibration curve, $R^2$, Brier score). A validation was performed on an external sample of 132 patients hospitalized in a French hospital close to Paris, in Aulnay-sous-Bois, Île-de-France. The probability of ICU transfer or death was 32% (47/147) (95% CI 25–40). Older age (OR 2.61, 95% CI 0.96–7.10), poorer respiratory presentation (OR 4.04 per 1-point increment on World Health Organization (WHO) clinical scale, 95% CI 1.76–9.25), higher CRP-level (OR 1.63 per 100mg/L increment, 95% CI 0.98–2.71) and lower lymphocytes count (OR 0.36 per 1000/mm³ increment, 95% CI 0.13–0.99) were associated with an increased risk of ICU requirement or death. A 9-point ordinal

the restrictions is ' comité d'éthique de la recherche, CER- Sorbonne Université. Researchers can contact Mr. Patrick Lefebvre with any inquiries. Mr. Patrick Lefebvre Email: Patrick.lefebvre@aphp.fr Tel: 0033631145043 Fax: 0033142162527 Address: DSI GH Pitié Salpetriere, 47-83 Bd de l'Hôpital, 75651 Paris cedex 13

**Funding:** The author(s) received no specific funding for this work. The funders had no role in study design, data collection and analysis, decision to publish, or preparation of the manuscript.

**Competing interests:** The authors have declared that no competing interests exist. This does not alter our adherence to PLOS ONE policies on sharing data and materials.

scale scoring system defined low (score 0–2), moderate (score 3–5), and high (score 6–8) risk patients, with predicted respectively 2%, 25% and 81% risk of ICU transfer or death at D14. Therefore, in this prospective cohort study of laboratory-confirmed COVID-19 patients hospitalized in a medical ward in France, a simplified scoring system at admission predicted the outcome at D14.

## Introduction

In January 2020, the World Health Organization (WHO) declared the outbreak of coronavirus disease 2019 (COVID-19) to be a Public Health Emergency of International Concern [1]. This outbreak started in China (Wuhan), from where most of the data is available to now. Clinical presentation varies widely among individuals. Although population-based data are lacking, up to one third of patients might be asymptomatic [2, 3]. Among the symptomatic ones, more than 80% develop a mild disease, while only a minority presents the severe form of severe acute respiratory syndrome coronavirus 2 (SARS-CoV2) infection [4]. Intensive care unit (ICU) admissions range from 5% to 16%, depending on characteristics of the studied population [5, 6]. Also, Chinese retrospective studies reported an inpatient mortality rate of 17.6–28.2%, with median time to death between 15 and 18.5 days [7, 8]. Different prognostic factors emerge in this context, such as age and comorbidities [9, 10]. After Asia, Europe was quickly and severely affected by the epidemic. First in Italy then in France, the outbreak rapidly overwhelmed the public health system and ICUs were filled. As of May 12th 2020, France had already confirmed 177.547 cases with 26.646 deaths [11].

Currently, there are no validated treatments for COVID-19 and huge efforts have allowed designing and implementing very rapidly randomized controlled trials. Also, predictive prognostic factors are critical to improve management of high-risk COVID-19 patients. It is crucial to early identify those at risk of worsening for (i) an optimized management of patients' flow and to (ii) to define the population to treat, ensuring healthcare quality [12]. At this time, very limited prospective data is available on outcome and prognostic factors of COVID-19 patients among European population. Our objective through this French single-center prospective cohort study of 152 COVID-19 patients was to develop and validate multivariable predictive models for the patient status at day 14, i.e. (i) major clinical worsening (death or ICU transfer by day 14), (ii) severe status at day 14 (remaining with non-invasive or mechanical ventilation, or death, at day 14), and (iii) favorable hospital outcome (discharge alive by day 14), in adult patients requiring initial hospitalization in a medical ward.

## Methods

### Study population

This is a prospective single-center observational cohort study of 152 COVID-19 adult patients admitted from March 16th 2020 till the 4th of April in the Internal Medicine and Clinical Immunology Department, at Pitié-Salpêtrière's Hospital, in Paris, France, a tertiary care university hospital. Included patients were those older than 18 years with initial requirement for hospitalization in medical ward, and diagnosed with COVID-19, defined as positive SARS-CoV-2 real-time reverse transcriptase–polymerase chain reaction (RT-PCR) assay from nasal swabs. Hospitalization criteria in medical ward was either the need for oxygen support (oxygen mask or non-invasive ventilation, but not mechanical ventilation) with hemodynamic stability,

or a high-risk comorbidity profile that would need close follow-up according to emergency room judgement.

All patients benefitted from current standard COVID-19 care at the time. The study followed the Strengthening Reporting of Observational Studies in Epidemiology (STROBE) and the TRIPOD reporting guideline for cohort studies [12] We received local ethical committee approval (Comité d'éthique de la recherche Sorbonne University, CER-2020-14), and our study is registered as (NCT04320017).

All data were prospectively collected in a standardized form from the medical files of the patients. At baseline (i.e., hospital admission), we assessed demography and epidemiology features, comorbidity profile, previous treatments, clinical presentation along with the laboratory, chest computed tomography (CT) scan and echocardiogram data. Routine blood examinations included full blood count, glycaemia, renal and liver function tests, creatine kinase, lactate dehydrogenase, C-reactive protein (CRP), procalcitonin, fibrinogen, D-dimer, troponin, ferritin and interleukin-6 (IL-6). CT scan imaging results were reported according to the predominant pattern of lesions and the extent of the lesions. The first administered treatments and clinical course during hospitalization were recorded.

Patients were categorized using the WHO clinical improvement Scale [13] on day 1 (D1) and day 14 (D14). This 9-point ordinal scale measures illness severity over time as follows: 0, uninfected; 1, ambulatory, no limitation of activities; 2, ambulatory, limitation of activities; 3, hospitalized, no oxygen therapy; 4, hospitalized, oxygen by mask or nasal prongs; 5, hospitalized, oxygen by non-invasive ventilation or high-flow; 6, intubation and mechanical ventilation; 7, ventilation with additional organ support (i.e., vasopressors, dialysis, extracorporeal membrane oxygenation); 8, death. All data were collected and reviewed by three physicians (AH, GM and MV). Patients discharged from hospital before D14 were contacted by phone to assess their status at that time point.

Eligibility criteria for the validation cohort was the same used for the development cohort, being carried out in another hospital close to Paris, in Aulnay-sous-Bois, Île-de-France. The outcome was defined and assessed in a similar way to that of development cohort. Data were collected from medical hospitalization records, which included the date of admission and, as appropriate, date of hospital discharge, date of ICU transfer, date of ICU discharge, date of invasive ventilation initiation and withdrawal, date of death. From those dates, outcomes at day 14 of admission were derived, as defined for the analyses.

Non opposition to participate was obtained from each participant, and a dated non opposition form was collected and included in their medical hospitalization records, following French legislation for observational studies on standard of care data.

## Definitions of study endpoints

The study endpoints were defined as the occurrence of ICU transfer or death within 14 days of admission (main endpoint), the need for non invasive or mechanical ventilation, or death, at day 14 after hospital admission, and being discharged alive within 14 days of admission.

## Statistical analysis

The sample size (number of individuals, $n = 152$) consisted in all consecutive eligible patients hospitalized at the study center, during the first weeks of the 2020 SARS-CoV2 outbreak in Paris, France. For descriptive analyses, categorical variables are reported with counts (percent) and quantitative variables with median [interquartile range]. The association between groups and variables was evaluated using Fisher's exact test for categorical variables, and with Wilcoxon's rank sum test for quantitative variables. Categorical variables were compared using

Fisher's exact test and quantitative variable with Wilcoxon's rank sum test. Analyses were performed on complete cases. Quantitative predictors were considered as continuous variables (except for age) and qualitative as binary or dummy variables, for model development. A set of predictors was defined after checking for redundancy among candidate predictors based on clinical expertise, as well as and multicollinearity, and accounting for an acceptable number of degrees of freedom given the limiting number of events. We considered predictors that would be available in most medical wards, in routine practice, representing patients status at baseline, both clinically and biologically. The predictor variables used were age, CRP level, lymphocyte count, and respiratory presentation presented as WHO score. These data are measured at the initial presentation of the patient. Poor respiratory presentation is defined as WHO score equal or superior to 5, oxygen by NVI or high flow oxygen (more than 6 L/min). No statistical-based variable selection was performed. The multivariable models of the endpoints of interest were evaluated using logistic regression models, with maximum likelihood. Validation was performed in two stages. Internal validation of the models was first performed using 1000 bootstrap resamples [14]; we estimated models performances, corrected for over-optimism (see S1 File). The models were further evaluated on an external validation sample from another French hospital close to Paris, in Aulnay-sous-Bois, Île-de-France (see S1 Table in S1 File). We defined a tentative simplified scoring system, for the main endpoint (ICU transfer or death within 14 days of admission); to that aim, continuous variables were to be dichotomized (for simplified field risk-assessment) and a unit coefficient was allocated to each of the model variables (see S1 File). The simplified score was validated internally using a resampling approach by bootstrap (number of bootstrap sample, N = 1000), and on the external cohort. For each variable, missing data was described with count. For model development, we used routinely obtained predictors (no missing data). All statistical tests were two-sided at a 5%-significance level. Analyses were performed on R statistical platform, version 3.5.3.

## Results

A total of 152 consecutive eligible patients were hospitalized in the ward and included in the study. The main baseline features are presented in Table 1. Median age was 77 years [60–83], male sex and Caucasian origin were predominant, and 80.9% of the patients had comorbidities. By the time of arrival, 28 (18.4%) patients reported angiotensin-converting-enzyme inhibitors as continuous-use medication, while 16 (10.5%) had taken nonsteroidal anti-inflammatory drugs. Dyspnea was the most frequently symptom, followed by fever and dry cough. On admission, 44 patients (28.9%) had a WHO score of 3, 89 patients (58.6%) had a WHO score of 4, and 19 patients (12.5%) had a score of 5. Half of the patients presented with lymphopenia, with values below 800 cells/mm$^3$. Chest CT scan showed that ground glass opacities were the most frequent lesions with an extent greater than 50% of the parenchyma evidenced in 24.7% of patients. IL-6 level was 31.8 pg/mL [14.8–56.0] and higher levels (161.1 pg/mL [32.7–237.8]) were observed in patients with extensive lung opacities (> 50%) as compared to those with a non-extensive lung involvement (31.7 pg/mL [15.4–51.6], p = 0.022). At admission, 129 (84.9%) patients received antibiotics, 68 (45%) hydroxychloroquine and 6 (3.9%) tocilizumab.

The study's flow-chart represents all patients' outcomes (Fig 1). Complete 14-day follow-up was available for 146 patients. During their clinical course, 56 (38.3%) patients experienced respiratory worsening, with 49 of them requiring an oxygen flow over 6 L/min at some point. As of day 14 (D14), 17 (11.6%) had been transferred to ICU, 5 to the semi-intensive unit, and, eventually, 32 (21.9%) patients had died, and 84 (57.5%) had been discharged alive from the hospital. For those who died, median time to death from symptom onset or hospital admission

**Table 1. Demographic, clinical, laboratory findings of patients and treatments on admission.**

| | All | ICU-free and alive | ICU or death | P value |
|---|---|---|---|---|
| Total sample‡ | 152 | 100 (68%) | 47 (32%) | |
| **Demographics** | | | | |
| Male patients | 91 (59.9%) | 59 (59%) | 31 (66%) | 0.47 |
| Age at admission (years) | | | | 0.014* |
| ≤ 60 | 41 (27%) | 34 (34%) | 7 (15%) | |
| 61–74 | 28 (18%) | 14 (14%) | 14 (30%) | |
| ≥ 75 | 83 (55%) | 52 (52%) | 26 (55%) | |
| Caucasian | 90/140 (64.3%) | 57/90 (63%) | 28/45 (62%) | |
| **Comorbidities** | | | | |
| Smoking | 10 (6.6%) | 9 (9%) | 0 (0%) | 0.058 |
| Hypertension | 82 (53.9%) | 52 (52%) | 25 (53%) | 1 |
| Diabetes | 37 (24.3%) | 25 (25%) | 12 (26%) | 1 |
| Dyslipidemia | 50 (32.9%) | 31 (31%) | 17 (36%) | 0.57 |
| Ischemic heart disease | 35 (23%) | 21 (21%) | 13 (28%) | 0.41 |
| Cancer | 30 (19.7%) | 20 (20%) | 9 (19%) | 1 |
| Chronic obstructive pulmonary disease | 12/151 (7.9%) | 7/99 (7%) | 4 (9%) | 0.75 |
| Ambulatory oxygen therapy | 3 (2%) | 0 (0%) | 3 (6%) | 0.031 |
| **Baseline on-going medications** | | | | |
| ACE inhibitor | 28 (18.4%) | 19 (19%) | 6 (12.8%) | 0.48 |
| NSAIDs | 16 (10.5%) | 12 (12%) | 4 (8.5%) | 0.78 |
| Corticosteroids | 16 (10.5%) | 11 (11%) | 5 (10.6%) | 1 |
| **Signs and symptoms on admission** | | | | |
| Days from first symptoms to admission | 5 (2;8) | 5 (2;9) | 5 (2;8) | 0.95 |
| Fever ≥ 38.8˚C | 38 (25%) | 23 (23%) | 13 (28%) | 0.54 |
| Respiratory rate ≥ 24 breaths per minute | 85/151 (56%) | 49 (49%) | 32/46 (70%) | 0.031 |
| SpO2 on room air, %† | 93 (90–96) | 94 (91–96)† | 91 (89–93) | 0.0001 |
| Oxygen therapy on admission | 110 (72.4%) | 65 (65%) | 42 (89%) | 0.003 |
| SpO2 on oxygen therapy, % | 96 (95–98) | 98 (95–99) | 95 (94–97) | 0.0009 |
| Oxygen flow, L/min | 2 (2–4) | 2 (2–3) | 3 (2–9) | 0.0008 |
| Anosmia | 17/150 (11.3%) | 13/99 (13%) | 3/46 (7%) | 0.39 |
| Dry cough | 68/151 (45%) | 43 (43%) | 23/46 (50%) | 0.48 |
| Dyspnea | 102/150 (67.5%) | 58 (58%) | 41/46 (89%) | 0.0001 |
| Myalgia | 32/150 (21.3%) | 27/99 (27%) | 5/46 (11%) | 0.031 |
| Fatigue | 70/150 (46.7%) | 50/99 (51%) | 20/46 (43%) | 0.48 |
| WHO clinical scale | 4 (3;4) | 4 (3;4) | 4 (4;5) | <0.0001 |
| **Laboratory findings** | | | | |
| Neutrophils, /mm³ | 4350 (2948–6962) | 4155 (2722–6145) | 5240 (3465–9120) | 0.020 |
| Eosinophils, /mm³ | 0 (0–22) | 10 (0–30) | 0 (0–10) | 0.014 |
| Lymphocytes, < 800/mm³ | 73 (48%) | 39 (39%) | 30 (64%) | 0.008 |
| C-Reactive protein, mg/L | 74.5 (30.9–135.1) | 56.6 (24.0–110.6) | 112.0 (66.2–212.9) | <0.0001 |
| Interleukine-6 | | | | 0.002 |
| ≥ 30 pg/mL | 31/55 (56.4%) | 17/38 (45%) | 13/14 (93%) | |
| Procalcitonin, ng/mL† | 0.2 (0.1–0.5) | 0.1 (0.1–0.3) | 0.4 (0.1–0.9) | 0.0001 |
| Ferritin, μg/L† | 913 (341–1612) | 786 (318–1348) | 1482 (758–2682) | 0.004 |
| Troponin, ng/mL† | 18.6 (9.4–39.7) | 16.5 (7.9–31.1) | 24.4 (14.2–47.7) | 0.020 |
| Lactate dehydrogenase, U/L† | 364 (284–444) | 349 (272–418) | 404 (311–498) | 0.044 |
| D-Dimer, μg/L† | 890 (570–1775) | 830 (510–1270) | 1550 (825–2305) | 0.022 |

*(Continued)*

**Table 1.** (Continued)

| **Imaging Studies** | | | | |
|---|---|---|---|---|
| No. | 105 | 70 | 32 | |
| Signs of SARS-CoV2 pneumonia | 101/103 (98%) | 66/68 (97%) | 32 (100%) | 1 |
| Stage | | | | 0.009* |
| No lesions | 2/103 (2%) | 2/68 (3%) | 0 (0%) | |
| Ground-glass opacity | 48/103 (47%) | 36/68 (53%) | 10 (31%) | |
| Consolidation | 36 /103(35%) | 24/68 (35%) | 11 (34%) | |
| Bilateral pulmonary infiltration | 17/103 (17%) | 6/68 (9%) | 11 (34%) | |
| More than 50% | 25/103 (24%) | 10/61 (15%) | 13 (41%) | 0.009 |
| **Echocardiograhy** | | | | |
| No. | 63 | 46 | 15 | |
| Left ventricular ejection fraction, %† | 65 (60–65) | 65 (65–65) | 65 (52–65) | 0.065 |
| **Medications received during hospitalization** | | | | |
| Hydroxychloroquine | 68 (45%) | 48 (48%) | 20 (43.5%) | 0.72 |
| Tocilizumab | 6 (4.1%) | 2 (2.1%) | 4 (8.7%) | 0.087 |

ACE, angiotension-converting enzyme; NSAIDs, nonsteroidal anti-inflammatory drugs; SARS-CoV-2, severe acute respiratory syndrome coronavirus 2; WHO, World Health Organization; SpO2, peripheral capillary oxygen saturation. Data are median (IQR), n (%) or n/N (%). P values were calculated by Mann-Whitney U test, Fisher's exact test, as appropriate.

‡5 of the 152 patients has incomplete follow-up for ICU transfer or death within 14 days

*Fisher's exact test comparing all subcategories.

†: number of missing values for quantitative variables: SpO2 $n$ = 13, Procalcitonine $n$ = 21, Ferritin $n$ = 59, Troponin $n$ = 22, LDH $n$ = 46, D-Dimer $n$ = 74, LVEF $n$ = 2 among patients with echocardiography [for categorical variables, in case of missing values, the denominator in the table indicates the number of complete cases].

were 11 (6.75–14.5) and 6 (4–9) days, respectively. The estimated probability of ICU transfer or death by D14 was 32% (95% CI 25–40), the estimated probability of still needing non-invasive ventilation (NIV) or mechanical ventilation (MV), or being dead, at D14 was 27% (95% CI 20–35), while the estimated probability of being discharged alive by D14 was 58% (95% CI 49–66).

In univariable analysis, age at admission, chronic respiratory failure, respiratory rate ≥ 24 breaths per minute, peripheral capillary oxygen saturation (SpO2) on room air, oxygen therapy on admission, SpO2 on oxygen, dyspnea, myalgia, WHO clinical scale, neutrophilia, eosinopenia, lymphopenia, CRP level, IL-6 level, procalcitonin, fibrinogen, serum ferritin, high-sensitivity cardiac troponin T, lactate dehydrogenase (LDH), D-dimer, and chest CT scan were associated with ICU transfer and/or death within 14 days (Table 1). For adjusted model development, the limiting number of events was 47 patients with ICU transfer or death within 14 days in the original sample. The multivariable model included age (≤ or > 60 years), respiratory baseline presentation (assessed by WHO scale levels from 3 to 5), CRP level and lymphocytes count. Older age (OR 2.61, 95% CI 0.96–7.10), poorer respiratory presentation (OR 4.04 per 1-point increment on WHO scale, 95% CI 1.76–9.25) and higher CRP level (OR 1.63 per 100mg/L increment, 95% CI 0.98–2.71) were associated with an increased risk of ICU requirement or death, while lymphocytes count were associated with better outcome (OR 0.36 per 1000/mm$^3$ increment, 95% CI 0.13–0.99) (Fig 2, S2 Table in S1 File). Fig 2 shows a forest plot of the multivariable models of COVID-19 patient's outcomes. Internal and external validation of the model was performed: the C-index (equivalent to AUC) was 0.80, 0.78 after correction for over-optimism by resampling, and 0.78 on the external cohort (see S1 File for further details and S1 Table in S1 File for description of the external cohort).

**Fig 1. Flow chart of COVID-19 patients' outcome.** Maximum follow-up = 14 days. *at the time of analysis, 6 patients were still followed up for the study endpoint, 5 hospitalized in the medical ward and 1 in the ICU.

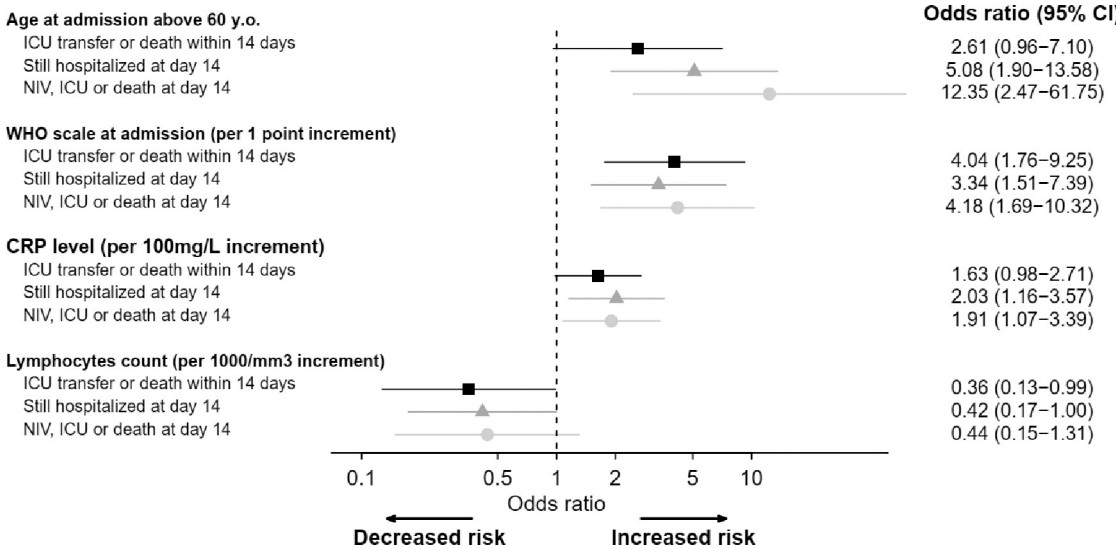

**Fig 2. Forest plot of multivariable analysis of COVID-19 patients' outcome [black squares: Model for ICU transfer or death within 14 days of admission (main endpoint, analysis on 147 observations); gray triangles: Model for hospitalization status at day 14 (analysis on 146 observations); gray circles: Model for detailed status at day 14 (analysis on 146 observations)].**

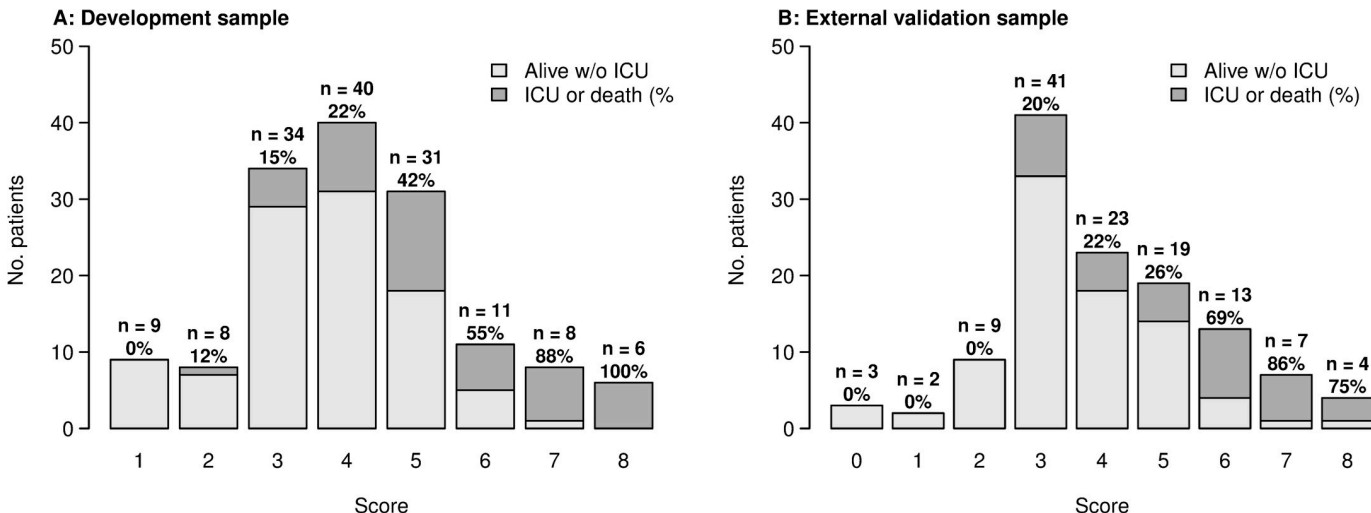

**Fig 3. Proportions of ICU transfer or death within 14 days after admission by risk score.** Left panel A: development cohort. Right panel B: external validation cohort.

A tentative simplified scoring system was defined for the main endpoint (ICU transfer or death within 14 days of admission), for routine clinical field practice. To that aim, based on the linear predictor and the coefficients of the multivariable model, in an additive manner, 1 point was allocated for age above 60 years old; 1 point for oxygen therapy by nasal prongs or mask (WHO scale level 4); 3 points for high flow oxygen or NIV (WHO scale level 5); 1 point if $10 \leq$ CRP plasma level $\leq 75$ mg/L, 2 points if $75 \leq$ CRP $\leq 150$ mg/L, 3 points if CRP $\geq 150$ mg/L; 1 point if lymphocytes count below 800/mm3 (See S1 File). Fig 3 displays stratified risk according to each score. Therefore, we defined three risk groups: low (score 0–2), moderate (score 3–5), and high (score 6–8). Cumulative incidence for each of these groups is shown in Fig 4. Overall, the estimated sensitivity of a score greater than 2 (moderate and severe risk groups) was 97% (95% CI 94–100), and the specificity of a score lower than 6 (low and moderate risk groups) was 94% (95% CI 89–98) for the main outcome. The positive predictive value for a high-risk score was 76% (95% CI 61–91), while the negative predictive value for a low risk score was 94% (95% CI 82–100).

At day 14, a total of 40 patients were still treated with NIV ($n = 1$) or MV ($n = 7$) ventilation, or had died ($n = 32$), out of 146 evaluable patients. In univariable analysis, age at admission, weight, chronic respiratory failure, respiratory rate $\geq 24$, SpO2 on room air, Oxygen therapy on admission, SpO2 on oxygen, dyspnea, myalgia, WHO clinical scale, neutrophils, eosinophils, lymphocytes, platelets, CRP level, IL-6 level, procalcitonin, serum ferritin, high-sensitivity cardiac troponin T, D-dimer, and chest CT-scan were associated with WHO scale $\geq 5$ within day 14. Multivariable analysis is represented in Fig 2.

Eighty-four patients had been discharged by day 14, out of 146 evaluable patients. In univariable analysis, age at admission, respiratory rate $< 24$, SpO2 on room air, Oxygen therapy on admission, ageusia, dyspnea, WHO clinical scale, neutrophils, eosinophils, lymphocytes, platelets, CRP level, IL-6 level, procalcitonin, fibrinogen, serum ferritin, high-sensitivity cardiac troponin T, LDH, D-dimer, and chest CT scan were associated with discharge alive within 14 days. Multivariable analysis is represented in Fig 2.

## Discussion

The natural history and outcome of the COVID-19 patients initially hospitalized in a medical ward remain unpredictable. Currently, the main existing medical information stem from

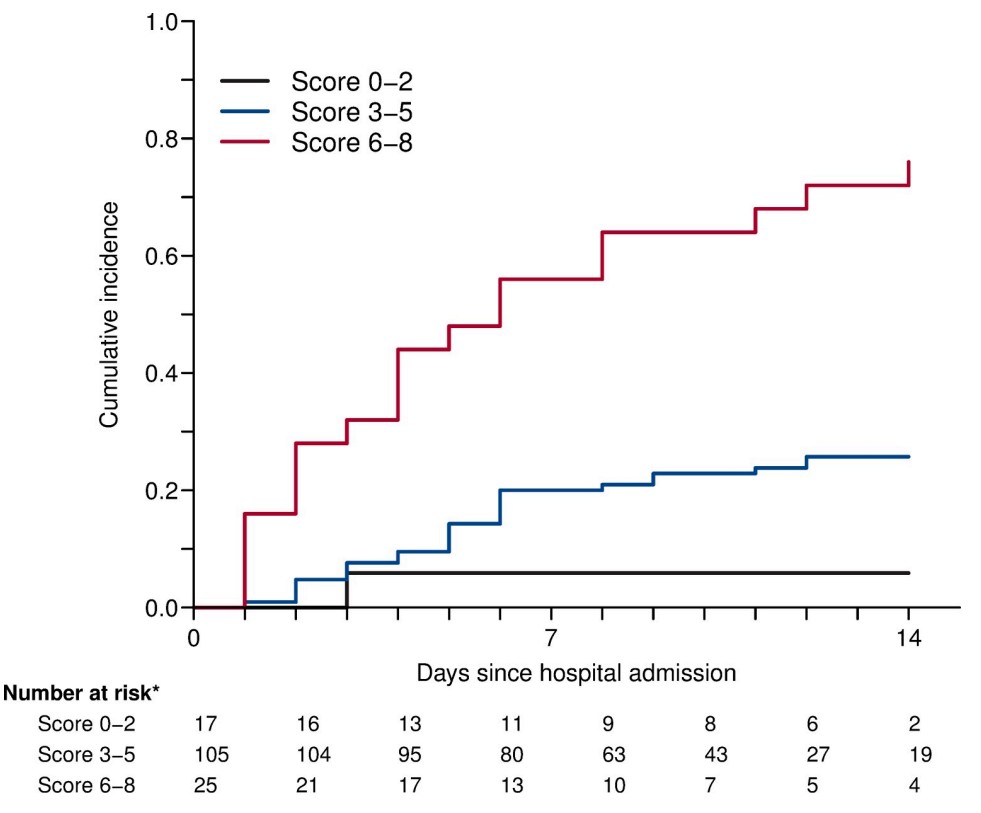

**Fig 4. Cumulative incidence of ICU transfer or death by risk score.**

China and prognostic factors of COVID-19 among European population are lacking. The most striking conclusions drawn by this study are (i) up to 35% of the COVID-19 patients hospitalized in a medical ward were transferred to ICU or died at day 14, (ii) we defined high-risk group of ICU transfer or death using a simplified scoring system from the multivariable models including age, CRP level, lymphocytes count and WHO scale and (iii) we highlighted correlation between IL-6 level and extensive lesions in CT scan.

A clear and strong age gradient in death risk has been identified, increasing dramatically after 60 years [15]. Besides older age, comorbidities are also highlighted as key factors associated with death [7, 8, 16]. Compared to the present study, retrospective Chinese cohorts population were younger (from 51 to 56 years) and had less comorbidities (up to 48%) [7, 16]. Even with a median age of 77 years and more than 80% of comorbidity, our reported 21.9% mortality rate lies within the 17.6–28.2% range extracted from other cohorts [7, 8]. In contrast, the median time from symptoms onset to death in our population (11 days) is shorter than the 18.5 days previously reported [7], which can be ultimately the consequence of the higher risk profile of patients in the present study. Additionally, our ICU transfer rate (11.6%) was lower than the 26% described in Chinese cohorts [7, 16]. In this regard, we must underline that our patients presented with less severe infection at baseline [7, 16]. In addition, they were less eligible to ICU admission, due to age and comorbidities. Beyond demographic and clinical characteristics, several laboratory features have been linked to a higher mortality. Studies identified a positive correlation with mortality for neutrophilia, lymphopenia, troponin, LDH and D-dimer levels [7, 16]. Additionally, high levels of serum CRP, procalcitonin, and ferritin have also occasionally been associated with mortality [16, 17]. In our cohort, two simple biomarkers from routine practice, lymphocytes count and CRP level, are independently associated with a

worse prognosis. CRP level higher than 75 mg/L and lymphopenia below 800/mm$^3$ increased by two fold the odds of being transfer in ICU or death.

Herein, we provided for the first time a simplified scoring system which allows stratifying COVID-19 patients initially hospitalized in a medical ward, at low, intermediate, or high risk of ICU transfer or death. The score was validated with calibration evaluated both with an internal resampling approach and by external validation on a cohort sample from a different hospital. Based on the linear predictor of the multivariate model, age above 60 years, WHO scale, CRP level (10–75, 75–150, or > 150 mg/L), and lymphocytes count below 800/mm3 were included in the scoring system. A score equal or greater than 6 at baseline had a predicted probability of more than 60% to be transferred to ICU or dead by D14. In our regard, this high-risk patient profile should be monitored more closely and eventually considered for more aggressive treatment protocols than a patient with a score of less than 3. In a systematic review of the prediction models for diagnosis and prognosis of COVID 19 patients, Wynants et al identified ten prognostic models proposed by different Chinese teams [12]. By the time of this article writing, all these models were only available in pre-print and had not been peer-reviewed. They were exclusively based on small retrospectives cohorts, with most of them lacking an external validation cohort, or presenting a non-comparable small validation cohort. Nguyen et al [18] developed a 7-point score based on a retrospective analysis of 279 hospitalized patients but without external validation. The strengths of the score presented here are its prospective nature and its external validation. In addition, its readily accessible variables make it easily reproducible in clinical practice.

Apart from CRP level and lymphocyte count, other significant findings from our study could be further used to refine the score. Chest CT scan is a useful diagnostic tool, especially for RT-PCR negative patients, but its role as a prognostic instrument is still unclear [19]. Herein, we pointed out that parenchymal involvement greater than 50% on chest CT scan at admission was associated with ICU transfer or death in 41% of cases. In parallel, high levels of serum IL-6 have been reported in moderate to severe cases of COVID-19 pneumonia [7, 17]. IL-6 may result in increased alveolar-capillary blood-gas exchange dysfunction, especially impaired oxygen diffusion, and lead to pulmonary fibrosis and organ failure [20]. We were able to establish for the first time the correlation between IL-6 level and extensive parenchymal involvement on chest CT scan for ICU transfer or death.

Our study has some limitations. We presented models with both internal and external validation. Discrimination of the model and of the simplified score for the main endpoint was consistent in the external cohort. Calibration assessment showed a slightly overestimated risk of event in the external cohort for those with higher scores. The external sample consisted of patients from a regional non-university hospital, which could explain the differences on catchment area and patient recruitment. In the acute context of the first SARS-CoV-2 epidemic wave in France, we relied on a sample prospectively defined by consecutive eligible patients in the study center. Overall, the limited sample sizes of both development and validation samples require caution in interpreting results. Ideally, a sample size calculation at planning stage of the study should ensure sufficient collected data for predictive model development and validation; approaches have been proposed to that aim [21, 22]. Further external validation on larger prospective cohorts with planned sample sizes will be useful.

To our knowledge, this is the first prospective European cohort of COVID-19 non-critical inpatients and one the largest standardized studies describing short term patients outcome. We provided a very simple and easily accessible score to estimate the risk of ICU transfer or death by day 14. In the context of the pandemic, this tool can help the management of patient flow, and also clinical trial design and therapeutic management.

## Supporting information

**S1 File. Supplementary material.**
(DOCX)

## Acknowledgments

DIMICOVID is Département de Médecine Interne et Immunologie clinique groupe pour COVID-19 à l'Hôpital Pitié-Salpêtrière, APHP, Paris Sorbonne includes: Yves Allenbach, David Saadoun, Georgina Maalouf, Matheus Vieira, Alexandra Hellio, Jacques Boddaert, Hélène Gros, Joe Elie Salem, Olivier Benveniste, Patrice Cacoub, Ahlem Chaib, Nicolas Champtiaux, Aude Rigolet, Anne Simon, Stéphane Barete, Jean-Charles Piette Perrine Guillaume-Jugnot, Yasmina Ferfar, Mathieu Vautier, Ségolène Toquet-Bouedec, Christian de Gennes, Fanny Domont, Gaëlle Leroux, Mathilde Leclercq, Chloé Comarmond, Anne-Claire Desbois, Nabiha Sbeih, Amine Ghembaza, Joana Alves-vieira, Hugues Gontier, Sofia Garabetyan, Marion Larue, Andréa Patissier, Elissone Sarkis, Sandrine Tramond, Roxana-Maria Bogdan, Nicias Gorge, Benjamin Rossi, Marie Anne Bouldouyre, Hélène Guillot, Keito Le Goff, Leila Lefevre, Serge Barmo, Ana-Maria Cardamisa, Margot Hulin, Alexandre Lejoncour, Céline Anquetil, Bailly Laurent, Corti Léonard, Gonçalo Boleto, Cindye Marques, Félix Blanc, Charlotte Bouzbib, Sara Philonenkon, Violaine Foltz, Jeremy Rezai, Christiane Stern, Manon Allaire, Philippe Sultanik, Oussama Mouri, Alessandra Mazzola, Frédérique Gandjbakhch, Eouard Larrey, Laure Gossec, Charlotte Tomeo, Vincent Mallet, Clémence Fron, Marika Rudler, Aline Lecleach, Bruno Fautrel, Pascal Lebray.

## Author Contributions

**Conceptualization:** Yves Allenbach, David Saadoun, Olivier Benveniste, Patrice Cacoub.

**Data curation:** Yves Allenbach, David Saadoun, Georgina Maalouf, Matheus Vieira, Alexandra Hellio, Jacques Boddaert, Hélène Gros, Joe Elie Salem.

**Formal analysis:** Lucie Biard.

**Investigation:** Yves Allenbach, David Saadoun, Georgina Maalouf, Matheus Vieira.

**Methodology:** Matthieu Resche Rigon, Cherifa Menyssa, Lucie Biard.

**Software:** Matthieu Resche Rigon, Cherifa Menyssa, Lucie Biard.

**Supervision:** Patrice Cacoub.

**Validation:** Olivier Benveniste.

**Writing – original draft:** Yves Allenbach, David Saadoun, Georgina Maalouf, Matheus Vieira.

**Writing – review & editing:** Jacques Boddaert, Matthieu Resche Rigon, Lucie Biard, Olivier Benveniste, Patrice Cacoub.

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
