## [Decision Letter · Decision Letter 0]

20 Jul 2020

PONE-D-20-14374

Multivariable prediction model of ICU transfer and death: a French prospective cohort study of COVID-19 patients.

PLOS ONE

Dear Dr. Maalouf,

Your submission has now been peer-reviewed by two experts in the field and myself. I agree that the manuscript would benefit from being revised according to the suggestions following and encourage you to do so.

I have read with great interest your manuscript and firmly believe that it contributes to a better understanding of the Covid situation. However, I should point out that several methodological flaws exist and would suggest revising accordingly.

Title:

I would emphasize the target population and change slightly to " *Development a multivariate prediction model of intensive care unit transfer or death: A French prospective cohort study of hospitalized COVID-19 patients*"

Abstract:

The setting where the study took place should be reported for both the development and validation datasets;Describe the statistical methods used, particularly the model calibration for both the developmental and validation datasets;Please describe the number of events (for the primary outcome) and their respective percentual proportion for both the development and validation datasets;

Methods

IRB protocol number is missing;Describe eligibility criteria for participants fo both the development and validation datasets;Describe the source of data for both the development and validation datasets. Identify any difference from the development data in setting, eligibility criteria, outcome, and predictors. Differences or similarities in definition with the development study should be described;Clearly define the primary outcome and the outcome that will be the target of analysis in the clinical risk score validation.Describe how the outcome was assessed in the validation cohortDescribe all the predictor variables used in developing or validating the multivariable prediction model, including how and when they were measured. What do you mean by poor respiratory presentation?Describe how missing data were handled with details of any imputation method used. If missing data were imputed, a description of which variables were included in the imputation procedure should be given. Could you mention the proportion of variables that had missing values?Could you specify all measures used to assess model performance in both developing and validation datasets?

Discussion

Discuss in your limitations, the inability to accurately estimate the sample size needed when developing a clinical prediction model for binary or time-to-vent outcomes. Ideally, the sample size should be large enough to minimize model overfitting and target sufficiently precise model predictions.

We look forward to receiving your revised manuscript.

Kind regards,

José Moreira, MD, MSc

Academic Editor

PLOS ONE

Journal Requirements:

'The study received local ethical committee approval, and is registered as (NCT04320017).'

(a) Please amend your current ethics statement to include the full name of the ethics committee/institutional review board(s) that approved your specific study.

(b) Once you have amended this/these statement(s) in the Methods section of the manuscript, please add the same text to the “Ethics Statement” field of the submission form (via “Edit Submission”).

For additional information about PLOS ONE ethical requirements for human subjects research, please refer to " ext-link-type="uri" xlink:type="simple">http://journals.plos.org/plosone/s/submission-guidelines#loc-human-subjects-research."

3. Please provide additional details regarding participant consent.

In the ethics statement in the Methods and online submission information, please ensure that you have specified (i) whether consent was informed and (ii) what type you obtained (for instance, written or verbal, and if verbal, how it was documented and witnessed).

If your study included minors, state whether you obtained consent from parents or guardians.

If the need for consent was waived by the ethics committee, please include this information.”

4. In your Methods section, please provide additional information about the participant recruitment method and the demographic details of your participants.

Please ensure you have provided sufficient details to replicate the analyses such as:

a) the recruitment date range (month and year),

b) a description of any inclusion/exclusion criteria that were applied to participant recruitment,

c) a description of how participants were recruited, and

d) descriptions of where participants were recruited and where the research took place.

5. Please include additional information regarding the survey or questionnaire used in the study and ensure that you have provided sufficient details that others could replicate the analyses.

For instance, if you developed a questionnaire as part of this study and it is not under a copyright more restrictive than CC-BY, please include a copy, in both the original language and English, as Supporting Information.

6. We note that you have indicated that data from this study are available upon request. PLOS only allows data to be available upon request if there are legal or ethical restrictions on sharing data publicly. For information on unacceptable data access restrictions, please see http://journals.plos.org/plosone/s/data-availability#loc-unacceptable-data-access-restrictions.

7. One of the noted authors is a group or consortium; DIMICOVID.

In addition to naming the author group, please list the individual authors and affiliations within this group in the acknowledgments section of your manuscript.

Please also indicate clearly a lead author for this group along with a contact email address.

8. Please amend either the abstract on the online submission form (via Edit Submission) or the abstract in the manuscript so that they are identical.

Reviewers' comments:

Reviewer's Responses to Questions

**Comments to the Author**

1. Is the manuscript technically sound, and do the data support the conclusions?

Reviewer #1: Partly

Reviewer #2: Yes

2. Has the statistical analysis been performed appropriately and rigorously? 

Reviewer #1: I Don't Know

Reviewer #2: Yes

3. Have the authors made all data underlying the findings in their manuscript fully available?

Reviewer #1: No

Reviewer #2: Yes

4. Is the manuscript presented in an intelligible fashion and written in standard English?

Reviewer #1: Yes

Reviewer #2: Yes

5. Review Comments to the Author

Reviewer #1: Thank you for the opportunity to review this work. I believe there is currently a great demand for prognostic models for COVID-19. The authors have developed a novel prediction score with a population of 152 adults with COVID-19. They conducted validation internally using bootstrapping, and with an external cohort of 131 patients recruited at another hospital. The authors list three outcomes of interest, but their main focus appears to be ICU admission or death within 14 days.

A key concern is the number of events described. In their development sample, 47 people died or were admitted to ICU, and in the validation sample 36. Collins et al. recommend at least 100 and ideally 200 to validate a prognostic model (https://doi.org/10.1002/sim.6787). It would be ideal if the authors were able to validate their score in a sample of this size. If not, the discussion should draw readers’ attention to this issue, and advise caution before this score is used.

I would like to see more details in the main manuscript on the methods used to develop the score. Particular issues are:

1. In univariable analyses, 21 variables are significantly related to ICU admission or death. How were the four included in the multivariable analysis chosen?

2. How were the variables in the multivariable model converted into the score? Of especial note is that CRP is responsible for 3 out of 8 possible points in the score but was statistically non-significant in the multivariable analysis for the outcome of ICU admission or death.

3. There are three different models described in the supplement for the three different outcomes. I think the manuscript needs to be clear what outcome the score is intended to predict.

4. For the calibration curves, it is unclear which model the predicted values come from.

Oxygen requirement, particularly non-invasive ventilation, is included as a risk factor and as a component in the prediction score, and is also treated as an outcome. While I appreciate that these are at different timepoints, I think this needs to be addressed in the discussion.

Could the authors provide more detail about the validation cohort please? Were data collected for this validation or for another purpose? Are there any differences between the cohorts (e.g. different size hospital, recruited at different time in the epidemic) or differences in data collection and follow-up? Any missing data in the validation cohort? It might help to add a second panel to Figure 1 with details of the validation cohort.

The methods state that missing data were described with a count, but these do not appear to be reported.

Figure 1. It would be clearer to have a box showing excluded patients, then the number included in the analysis, rather than the footnote. There are 6 patients who do not have 14 days of follow-up. The one admitted to ICU appears to be included in the analysis but the other five not.

Figure 2. The caption for Figure 2 describes a forest plot, but no forest plot is included in the PDF.

Figure 4 shows time-to-event details, but no time-to-event analyses are discussed. I think it would be clearer to show the calibration plots here, or possibly a receiver operating characteristic curve.

The discussion references other prognostic scores for COVID-19. It would help readers to describe how this novel score compares to those. It would also be useful to see how the authors anticipate the score being used.

They have identified three risk levels. It would be useful to see positive- and negative-predictive values for these thresholds. I would also like more details in the discussion of how the authors anticipate that these levels should be used (keeping in mind the initial concern regarding validation with a larger dataset).

Minor comments:

1. The WHO scale described is a 9-point scale, not 8.

2. The text on page 8 states ‘more than half showing values below 800 cells’, but the table gives this proportion as 48%

3. Typo in Table 1 ventricule for ventricle

4. Typos on page 14: presente for present (twice)

5. Typo halfway down page 15 ‘probability N of’

6. Typo of VNI for NIV in the supplementary table

7. Could the authors please define ‘high flow oxygen’ as a flow rate so this is unambiguous for international readers who might want to apply the score.

Reviewer #2: Major comments

1. It is well-known that, in prognostic cohort studies, it is important to have an adequate sample size when developing a prediction model. A larger sample size will yield more findings of high reliability. Although what constitutes an “adequate” sample size is a difficult question, some results there exist in the literature. From this point of view the authors should verify if their study meets the rule(s) of thumb suggested in:

-- Peduzzi P, Concato J, Kemper E, Holford TR, Feinstein AR. A simulation study of the number of events per variable in logistic regression analysis. J Clin Epidemiol 1996; 49:1373-1379

2. The authors should discuss the important multicollinearity issue.

3. The authors say that missing data were described with count, but the way they are dealt with in the regression analysis is unclear. The authors should discuss this aspect.

4. In addition to calibration and discrimination, the authors should evaluate the strength of the predictions from the model. In these terms, measures of performance (not of association) are needed. For binary outcomes, recently proposed R2 or Brier score can be used to present overall performance measures; see

-- Nagelkerke NJ. A note on a general definition of the coefficient of determination. Biometrika 1991; 78:691-692

-- Tjur T. Coefficients of determination in logistic regression models—A new proposal: the coefficient of discrimination. Am Stat 2009; 63;366-372

-- Rufibach K. Use of Brier score to assess binary predictions. J Clin Epidemiol 2010; 63:938-939

Minor comments

1. The symbol “n” for the sample size is used but never defined. Moreover, it is not in italic. Please check.

2. The authors write: “Categorical variables were compared using Fisher’s exact test and quantitative variable with Wilcoxon’s rank sum test.” I think it should be more correct to say that: “The association/dependence between variables was evaluated using Fisher’s exact test for categorical variables, and with Wilcoxon’s rank sum test for quantitative variables.”

6. PLOS authors have the option to publish the peer review history of their article (what does this mean?). If published, this will include your full peer review and any attached files.

Reviewer #1: No

Reviewer #2: No

---

## [Author Response · Author response to Decision Letter 0]

23 Sep 2020

Dear Editor and reviewers, 

We thank you for your valuable comments and advices on our manuscript “PONE-D-20-14374, ‘‘Multivariable prediction model of ICU transfer and death: a French prospective cohort study of COVID-19 patients.” 

You will find in this letter our response, point by point, to your comments and the revised changes that were done on the manuscript according to your recommendations. 

We hope that we adequately addressed all the required changes.

We stay at your disposal for any further necessary modifications. 

Yours Truly, 

Dr Georgina Maalouf 

Editor comments: 

Title:

• I would emphasize the target population and change slightly to " Development a multivariate prediction model of intensive care unit transfer or death: A French prospective cohort study of hospitalized COVID-19 patients”

The title has been revised following your suggestion. 

Abstract:

• The setting where the study took place should be reported for both the development and validation datasets

This information has been added in the revised abstract.

• Describe the statistical methods used, particularly the model calibration for both the developmental and validation datasets;

Statistical methods have been detailed in the abstract according to the reviewer’s recommendation.

• Please describe the number of events (for the primary outcome) and their respective percentual proportion for both the development and validation datasets; 

This information has been added to the revised abstract.

Methods

• IRB protocol number is missing

The IRB protocol number is CER-2020-14. 

The IRB number and the name of the ethics comity “Comité d’éthique de la recherche Sorbonne University “ were added to the Methods section.

• Describe eligibility criteria for participants fo both the development and validation datasets;

In the study population, we defined the eligibility criteria as follows: 

‘Included patients were those older than 18 years with initial requirement for hospitalization in medical ward, and diagnosed with COVID-19, defined as positive SARS-CoV-2 real-time reverse transcriptase–polymerase chain reaction (RT-PCR) assay from nasal swabs.’ 

We added: Hospitalization criteria in medical ward was either the need for oxygen support (oxygen mask or non-invasive ventilation, but not mechanical ventilation) with hemodynamic stability, or a high-risk comorbidity profile that would need close follow-up according to emergency room judgement.

Eligibility criteria for the validation cohort was the same as the development cohort. 

This information has been added in the revised manuscript to the Methods section.

• Describe the source of data for both the development and validation datasets.

Data were collected from the digitalized medical files and record of the patients in each hospital . 

This information has been added in the revised abstract to the Methods section, P7.

• Identify any difference from the development data in setting, eligibility criteria, outcome, and predictors. Differences or similarities in definition with the development study should be described;

Eligibility criteria for the validation cohort was the same used for the development cohort, being carried out in another hospital close to Paris, in Aulnay-sous-Bois, Île-de-France. The outcome was defined and assessed in a similar way to that of development cohort.

This information has been added in the revised manuscript to the Methods section, P8.

 Furthermore, a description of patients’ demographics, outcomes and predictors is provided in Supplementary Materials, table S1. Patients in the external validation cohort were younger (p0.0001) and had overall greater baseline lymphocytes counts (p=0.0003). There were no other significant differences in outcomes. It illustrates the difference in catchment area of the two centers. 

• Clearly define the primary outcome and the outcome that will be the target of analysis in the clinical risk score validation.

This information has been added in the revised manuscript to the Methods section, P9:

In the definition of study endpoints in the methods, we added main endpoint after “death or ICU transfer at Day 14”. 

In the Statistical Analysis, P10, we added : We defined a tentative simplified scoring system, for the main endpoint (ICU transfer or death within 14 days of admission).

• Describe how the outcome was assessed in the validation cohort 

The outcome was defined and assessed in the validation cohort similarly to the development cohort. Patients’ outcomes were collected from medical hospitalization records, which included the date of admission and, as appropriate, date of hospital discharge, date of ICU transfer, date of ICU discharge, date of invasive ventilation initiation and withdrawal, date of death. From those dates, outcomes at day 14 of admission were derived, as defined for the analyses. 

This information has been added in the revised manuscript to the Methods section, P8.

• Describe all the predictor variables used in developing or validating the multivariable prediction model, including how and when they were measured. What do you mean by poor respiratory presentation?

The predictor variables used in developing and validating the multivariable prediction model were age, CRP level, lymphocyte count, and respiratory presentation presented as WHO score. These data are measured at the initial presentation of the patient. Poor respiratory presentation is defined as WHO score equal or superior to 5, oxygen by NVI or high flow oxygen (more than 6 L/min). 

This information has been added in the revised manuscript to the Methods section, P10.

• Describe how missing data were handled with details of any imputation method used. If missing data were imputed, a description of which variables were included in the imputation procedure should be given. Could you mention the proportion of variables that had missing values?

Analyses were performed on complete cases samples. Sample sizes for each analysis are indicated in table 1 (in case of missing data: number of complete cases for qualitative variables is mentioned; for quantitative variables, number of missing data is stated in the table footnote). The sample size used for the prognostic model has been clarified and is now available in the title of figure 2 [n=147 for the primary endpoint, n=146 for hospitalization status at day 14]. Of note, variables with more than 20% missing data were not considered for the prognostic model (eg IL6, LDH, ferritin, etc.). We considered that variables which could not be easily and routinely collected at baseline were not relevant to elaborate the prognostic model and score.

• Could you specify all measures used to assess model performance in both developing and validation datasets?

For all outcomes, models performances were assessed with the following procedures, with internal (for over-optimism correction using 1000-boostrap resampling on the development dataset) and external validation: Brier score, for discrimination: Somers’ Dxy (C-index), R², for calibration: calibration intercept and slope (shrinkage) with estimation of the calibration curve via non parametric loess smoothing, and estimation of mean absolute calibration error. These aspects are available and detailed, with obtained results, in the Supplemental material.

Discussion

• Discuss in your limitations, the inability to accurately estimate the sample size needed when developing a clinical prediction model for binary or time-to-vent outcomes. Ideally, the sample size should be large enough to minimize model overfitting and target sufficiently precise model predictions.

We agree with the reviewer that sample size is an important aspect in developing predictive models. We have modified the discussion section accordingly to discuss this aspect in the limitations, P20. 

The study was conducted in the acute context of the first SARS-CoV-2 epidemic wave in France; we relied on a sample prospectively defined by consecutive eligible patients during the first weeks of COVID-19 activity in the study center. However, ideally, a sample size calculation at planning stage of a study should ensure sufficient information is collected for predictive model development and validation; approaches have been proposed to that aim [1-2]. More precisely, sample size (accounting for the number of events in our cases of binary endpoints) should be set to minimize model overfitting and target sufficiently precise estimation of the risk of event [2]. Such approaches should be used in planning and conducting further analyses for COVID-19 prognosis.

[1] Riley RD, Snell KIE, Burke D et al. Minimum sample size for developing a multivariable prediction model: Part I – Continuous outcomes. Statistics in Medicine 2019;38 :1262-75

[2] Riley RD, Snell KIE, Ensor J et al. Minimum sample size for developing a multivariable prediction model: Part I – binary and time-to-event outcomes. Statistics in Medicine 2019;38 :1276-96

Journal Requirements:

We have addressed these aspects in the revised submission.

'The study received local ethical committee approval, and is registered as (NCT04320017).' 

(a) Please amend your current ethics statement to include the full name of the ethics committee/institutional review board(s) that approved your specific study. 

The full name of the committee is Comité d’éthique de la recherche Sorbonne University.

The IRB number is “CER-2020-14”.

(b) Once you have amended this/these statement(s) in the Methods section of the manuscript, please add the same text to the “Ethics Statement” field of the submission form (via “Edit Submission”). 

This has been added accordingly in the Methods section P7 and in the submission form. 

3. Please provide additional details regarding participant consent. 

In the ethics statement in the Methods and online submission information, please ensure that you have specified (i) whether consent was informed and (ii) what type you obtained (for instance, written or verbal, and if verbal, how it was documented and witnessed). 

Non opposition to participate was obtained from each participant, and a dated non opposition form was collected and included in their medical hospitalization record, following French legislation for observational studies on standard of care data. This information has been added in the revised manuscript to the Methods section. 

We updated our statement on P 23 and in the submission form 

4. In your Methods section, please provide additional information about the participant recruitment method and the demographic details of your participants. Please ensure you have provided sufficient details to replicate the analyses such as:

a) the recruitment date range (month and year), 

The patient were included from March 16th 2020 until the 4th of April 2020. This has been added to the Methods section P 7. 

b) a description of any inclusion/exclusion criteria that were applied to participant recruitment, 

The inclusion criteria are detailed in the study population in the methods section, on page 7. 

“Included patients were those older than 18 years with initial requirement for hospitalization in medical ward, and diagnosed with COVID-19, defined as positive SARS-CoV-2 real-time reverse transcriptase–polymerase chain reaction (RT-PCR) assay from nasal swabs. Hospitalization criteria in medical ward was either the need for oxygen support (oxygen mask or non-invasive ventilation, but not mechanical ventilation) with hemodynamic stability, or a high-risk comorbidity profile that would need close follow-up according to emergency room judgement”

c) a description of how participants were recruited, and where the research took place.

Patient included were those consecutively admitted to the department of Internal Medicine and Clinical Immunology Department, at Pitié-Salpêtrière’s Hospital, in Paris, France, a tertiary care university hospital., during the inclusion period and who were eligible for the study and unopposed to participating. 

This information is available in the study population in the methods section, P7

5. Please include additional information regarding the survey or questionnaire used in the study and ensure that you have provided sufficient details that others could replicate the analyses. 

There was no specific survey or questionnaire used in the study. 

6. We note that you have indicated that data from this study are available upon request. PLOS only allows data to be available upon request if there are legal or ethical restrictions on sharing data publicly. For information on unacceptable data access restrictions, please see http://journals.plos.org/plosone/s/data-availability#loc-unacceptable-data-access-restrictions. 

We apologize, but, after double checking with the institution, unfortunately, data from this study cannot be made available (either publicly or upon request), for ethical and legal restrictions (data contain potentially indirectly identifying or sensitive patient information as per French regulation). 

This information was updated in our manuscript on P23 and in the submission form. 

7. One of the noted authors is a group or consortium; DIMICOVID. 

In addition to naming the author group, please list the individual authors and affiliations within this group in the acknowledgments section of your manuscript. 

DIMICOVID is the department of internal medicine and clinical immunology that managed COVID -19 patient in the Pitié-Salpêtrière Hospital , APHP, Paris Sorbonne. All the doctors that worked in the department during this period are cited in the acknowledgments on page 22. 

Please also indicate clearly a lead author for this group along with a contact email address. 

Yves Allenbach MD, yves.allenbach@aphp.fr

8. Please amend either the abstract on the online submission form (via Edit Submission) or the abstract in the manuscript so that they are identical. 

Both abstracts were updated with the revised version . 

Reviewer #1: 

Thank you for the opportunity to review this work. I believe there is currently a great demand for prognostic models for COVID-19. The authors have developed a novel prediction score with a population of 152 adults with COVID-19. They conducted validation internally using bootstrapping, and with an external cohort of 131 patients recruited at another hospital. The authors list three outcomes of interest, but their main focus appears to be ICU admission or death within 14 days. 

A key concern is the number of events described. In their development sample, 47 people died or were admitted to ICU, and in the validation sample 36. Collins et al. recommend at least 100 and ideally 200 to validate a prognostic model (https://doi.org/10.1002/sim.6787). It would be ideal if the authors were able to validate their score in a sample of this size. If not, the discussion should draw readers’ attention to this issue, and advise caution before this score is used. 

I would like to see more details in the main manuscript on the methods used to develop the score. Particular issues are:

1. In univariable analyses, 21 variables are significantly related to ICU admission or death. How were the four included in the multivariable analysis chosen?

As outlined by the reviewer, the main endpoint was ICU transfer or death within 14 days from admission. Given the limited sample size and distribution for this endpoint of interest (47 ICU transfer or death vs 100 alive without ICU transfer), we chose to allow on 4 degrees of freedom in the model predictor. We chose to spend these 4 degrees of freedom including clinically relevant variables and avoiding the unreliability of selection procedures.

We first evaluate all candidate variables for redundancy and collinearity, to reduce the candidate set. We also restricted to variables with limited missing values, aiming to consider only variables that are routinely collected and available upon admission, for external validity. Then we chose to include variables summarizing relevant information at admission; age (as it had been identified as a relevant prognostic factor for COVID-19), the baseline respiratory status, biological inflammation (CRP) and lymphocytes.

These aspects are reported in the supplementary material, section 2. 

2. How were the variables in the multivariable model converted into the score? Of especial note is that CRP is responsible for 3 out of 8 possible points in the score but was statistically non-significant in the multivariable analysis for the outcome of ICU admission or death.

We computed the tentative simplified score based on the linear predictor and the coefficients of the main multivariable model. Unit points were allocated as follows: 

- 1 point for age above 60 y.o.;

- 1 point for WHO scale at 4, 3 points for WHO scale 5; 

- CRP level was treated as: 

o 1 point if 10 ≤ CRP ≤ 75 mg/L, 

o 2 points if 75 ≤ CRP ≤ 150 mg/L, 

o 3 points if CRP ≥ 150 mg/L; 

- 1 point if lymphocytes count below 800/mm3. 

In the multivariate model, CRP is included as a continuous variable. Although it was not statistically significant at the 5%-level for the main endpoint, the estimate of CRP association with the endpoint, log(HR), is greater than 0, with a 95% confidence interval clearly asymmetrical around 0. This was consistent across the 3 endpoints we examined.

3. There are three different models described in the supplement for the three different outcomes. I think the manuscript needs to be clear what outcome the score is intended to predict.

We thank the reviewer for pointing this out. The manuscript has been modified to make that clearer, in the main text and in the supplementary material.

4. For the calibration curves, it is unclear which model the predicted values come from.

This has been clarified in the figure’s legends in the supplementary material

Oxygen requirement, particularly non-invasive ventilation, is included as a risk factor and as a component in the prediction score, and is also treated as an outcome. While I appreciate that these are at different timepoints, I think this needs to be addressed in the discussion. 

We thank the reviewer for raising this point. The population of the study was patients hospitalized for COVID-19 management in medical wards, excluding patients who were directly hospitalized in intensive care. Oxygen requirement is directly involved in the third endpoint: NIV, ICU or death at day 14. We considered baseline (admission=day 0) respiratory status as predictor for the outcome at day 14. We found that 15/18 patients with NIV at baseline were still receiving NIV at day 14. The initial need for NIV upon admission possibly represented a marker of disease severity and risk of prolonged need for respiratory support. 

Could the authors provide more detail about the validation cohort please? Were data collected for this validation or for another purpose? Are there any differences between the cohorts (e.g. different size hospital, recruited at different time in the epidemic) or differences in data collection and follow-up? Any missing data in the validation cohort? It might help to add a second panel to Figure 1 with details of the validation cohort. 

Data was collected specifically for this validation from patients hospitalization records. They were collected at the same time of the epidemic, during the first wave in Paris region, France. Furthermore, a description of patients demographics, outcomes and predictors is provided in Supplementary Materials, table S1. Patients in the external validation cohort were younger (p0.0001) and had overall greater baseline lymphocytes counts (p=0.0003). There was no other significant differences in outcomes. It illustrates the difference in catchment area of the two centers.

The methods state that missing data were described with a count, but these do not appear to be reported. 

Analyses were performed on complete cases samples. Sample sizes for each analysis are indicated in table 1 (in case of missing data: number of complete cases for qualitative variables is mentioned in the table as denominator for the percentage computation; for quantitative variables, the number of missing data, if any, is now stated in the table footnote).

Figure 1. It would be clearer to have a box showing excluded patients, then the number included in the analysis, rather than the footnote. There are 6 patients who do not have 14 days of follow-up. The one admitted to ICU appears to be included in the analysis but the other five not. 

Figure 1 has been modified to detail excluded patients excluded from analyses due to incomplete information in D14 status.

Figure 2. The caption for Figure 2 describes a forest plot, but no forest plot is included in the PDF. 

We apologize for the missing forest plot. It is included in the submitted revision documents, should now be visible, and is referred to in the manuscript.

Figure 4 shows time-to-event details, but no time-to-event analyses are discussed. I think it would be clearer to show the calibration plots here, or possibly a receiver operating characteristic curve. 

We agree with the reviewer that figure 4 relies on a time-to-event approach to represent patients’ outcome, whereas main analyses and models considered the outcome as a binary endpoint. Nevertheless, the cumulative incidences were computed using a competing event framework, with “discharge alive” and “ICU or death” as competing events; therefore these estimates are consistent with the proportions for the binary endpoint at day 14. A barplot for direct illustration of the binary endpoint is available in figure 3. 

Of note, as indicated in the supplementary material, a sensitivity analysis using the Fine and Gray model accounting for the time to the event (ICU transfer or death between day 1 and 14, with discharge alive without ICU stay as a competing event; discharge alive between day 1 and 14, with hospital death as a competing event) yielded similar results on the association of the variables with the outcomes distribution.

The discussion references other prognostic scores for COVID-19. It would help readers to describe how this novel score compares to those. It would also be useful to see how the authors anticipate the score being used. 

We agree with the reviewer on this point , we added in the discussion on P20 a comparison with another prognostic score to highlight the power of our score : 

Nguyen et al [20] developed a 7-point score based on a retrospective analysis of 279 hospitalized patients but without external validation. The strengths of the score presented here are its prospective nature and its external validation. In addition, its readily accessible variables make it easily reproducible in clinical practice.

Furthermore , we added in the discussion on P19 , an example of how the score can be used: 

“A score equal or greater than 6 at baseline had a predicted probability of more than 60% to be transferred to ICU or dead by D14. In our regard, this high-risk patient profile should be monitored more closely and eventually considered for more aggressive treatment protocols than a patient with a score of less than 3.”

1. Yann Nguyen, Félix Corre, Vasco Honsel, Vinciane Rebours, Bruno Fantin Adrien Galy, et.al A nomogram to predict the risk of unfavourable outcome in COVID-19: a retrospective cohort of 279 hospitalized patients in Paris area, Annals of Medicine, 52:7, 367-375,

They have identified three risk levels. It would be useful to see positive- and negative-predictive values for these thresholds.

Assuming the included sample is representative of the prevalence of ICU transfer or death in the population of interest (estimated prevalence 32%), the estimated PPV and NPV for the thresholds defining the 3 risk levels are reported in the table below, for each risk group, as well as Sensitivity and Specificity, with their 95% confidence intervals estimated by bootstrap resampling.

 Sensitivity Specificity PPV NPV

Score ≥ 3 97% (94;100) 16% (9;24) 35% (33;38) 94% (82;100)

Score ≥ 6 40% (28;55) 94% (89;98) 76% (61;91) 77% (73;82)

These estimates should be interpreted with caution since the sample corresponds to a sample included during the beginning of the first wave of SARS-CoV2 epidemics in Paris, France. The manuscript, in the discussion section, has been revised on these aspects.

I would also like more details in the discussion of how the authors anticipate that these levels should be used (keeping in mind the initial concern regarding validation with a larger dataset). 

We agree with the reviewer and this point was discussed with more details in the revised manuscript on P19: “A score equal or greater than 6 at baseline had a predicted probability of more than 60% to be transferred to ICU or dead by D14. In our regard, this high-risk patient profile should be monitored more closely and eventually considered for more aggressive treatment protocols than a patient with a score of less than 3.”

Minor comments:

1. The WHO scale described is a 9-point scale, not 8. We thank the reviewer for pointing out this issue. We have corrected the manuscript accordingly.

2. The text on page 8 states ‘more than half showing values below 800 cells’, but the table gives this proportion as 48% We thank the reviewer for pointing out this issue. We have corrected the manuscript accordingly. 

3. Typo in Table 1 ventricule for ventricle, We thank the reviewer for pointing out this issue. We have corrected the manuscript accordingly. 

4. Typos on page 14: presente for present (twice) We thank the reviewer for pointing out this issue. We have corrected the manuscript accordingly. 

5. Typo halfway down page 15 ‘probability N of’ We thank the reviewer for pointing out this issue. We have corrected the manuscript accordingly. 

6. Typo of VNI for NIV in the supplementary table of’ We thank the reviewer for pointing out this issue. We have corrected the manuscript accordingly. 

7. Could the authors please define ‘high flow oxygen’ as a flow rate so this is unambiguous for international readers who might want to apply the score. of’ We thank the reviewer for pointing out this issue. High flow oxygen was defined in statistical analysis on P6 : high flow oxygen (more than 6 L/min). 

Reviewer #2: Major comments 

1. It is well-known that, in prognostic cohort studies, it is important to have an adequate sample size when developing a prediction model. A larger sample size will yield more findings of high reliability. Although what constitutes an “adequate” sample size is a difficult question, some results there exist in the literature. From this point of view the authors should verify if their study meets the rule(s) of thumb suggested in:

-- Peduzzi P, Concato J, Kemper E, Holford TR, Feinstein AR. A simulation study of the number of events per variable in logistic regression analysis. J Clin Epidemiol 1996; 49:1373-1379

We agree with the reviewer that sample size is an important aspect in developing predictive models. 

The study was conducted in the acute context of the first SARS-CoV-2 epidemic wave in France; we relied on a sample prospectively defined by consecutive eligible patients during the first weeks of COVID-19 activity in the study center. Keeping that in mind and given the size of the obtained sample and number of events for the main analysis (47 out of 147), we chose to allow only 4 degrees of freedom to our main model and avoid variables selection procedures (beside clinical relevance, redundancy, collinearity) [1]. Indeed, ideally, a sample size calculation at planning stage of a study should ensure sufficient information is collected for predictive model development and validation; approaches have been proposed to that aim [2-3]. More precisely, sample size (accounting for the number of events in our cases of binary endpoints) should be set to minimize model overfitting and target sufficiently precise estimation of the risk of event [3]. Such approaches should be used in planning and conducting further analyses for COVID-19 prognosis. 

We have modified the discussion section accordingly to discuss this aspect in the limitations on page 20. 

[1] Harrell FR Jr. Regression Modeling Strategies. Second Edition, Springer International Publishing Switzerland 2015

[2] Riley RD, Snell KIE, Burke D et al. Minimum sample size for developing a multivariable prediction model: Part I – Continuous outcomes. Statistics in Medicine 2019;38 :1262-75

[3] Riley RD, Snell KIE, Ensor J et al. Minimum sample size for developing a multivariable prediction model: Part I – binary and time-to-event outcomes. Statistics in Medicine 2019;38 :1276-96

2. The authors should discuss the important multicollinearity issue.

We thank the reviewer for raising this point. For each endpoint, potential collinearity among the predictor variables was examined using a correlation matrix. There was no evidence of major multicollinearity among the predictors (all estimated correlation coefficient lower than 0.5, see table below). This has been added in the supplementatry material.

 Age WHO scale CRP

Age 

WHO scale 0.097 

CRP 0.020 0.456 

Lymphocytes count -0.137 -0.193 -0.239

3. The authors say that missing data were described with count, but the way they are dealt with in the regression analysis is unclear. The authors should discuss this aspect.

Analyses were performed on complete cases samples. Sample sizes for each analysis are indicated in table 1 (in case of missing data : number of complete cases for qualitative variables is mentioned; for quantitative variables, number of missing data is stated in the table footnote). The sample size used for the prognostic model has been clarified and is now available in the title of figure 2 [n=147 for the primary endpoint, n=146 for hospitalization status and detailed status at day 14]. Of note, variables with more than 20% missing data were not considered for the pronostic model (eg IL6, LDH, ferritin, etc.). We considered that variables which could not be easily and routinely collected at baseline were not relevant to elaborate the prognostic model and score.

These aspects have been clarified in the manuscript and the supplementary material.

4. In addition to calibration and discrimination, the authors should evaluate the strength of the predictions from the model. In these terms, measures of performance (not of association) are needed. For binary outcomes, recently proposed R2 or Brier score can be used to present overall performance measures; see

-- Nagelkerke NJ. A note on a general definition of the coefficient of determination. Biometrika 1991; 78:691-692

-- Tjur T. Coefficients of determination in logistic regression models—A new proposal: the coefficient of discrimination. Am Stat 2009; 63;366-372

-- Rufibach K. Use of Brier score to assess binary predictions. J Clin Epidemiol 2010; 63:938-939 

We agree with the reviewer and the Nagelkerke R² and Brier score for all models are available in Supplementary material, table 2.

Minor comments 

1. The symbol “n” for the sample size is used but never defined. Moreover, it is not in italic. Please check.

The notation “n” is now defined in the Statistical Analysis section and is typed in italic.

2. The authors write: “Categorical variables were compared using Fisher’s exact test and quantitative variable with Wilcoxon’s rank sum test.” I think it should be more correct to say that: “The association/dependence between variables was evaluated using Fisher’s exact test for categorical variables, and with Wilcoxon’s rank sum test for quantitative variables.” 

We have revised the sentence following the reviewer recommendation.

6. PLOS authors have the option to publish the peer review history of their article (what does this mean?). If published, this will include your full peer review and any attached files.

The authors prefer not to publish the peer review history.

---

## [Editor Report · Decision Letter 1]

2 Oct 2020

Development of a multivariate prediction model of intensive care unit transfer or death: A French prospective cohort study of hospitalized COVID-19 patients

PONE-D-20-14374R1

Dear Dr. Maalouf,

Thank you for submitting your article to PLoS One, which I have pleasure in accepting for publication as an original research paper. Well done and many congratulations! Your manuscript has now been formally accepted for publication. Please see important details concerning the publication process below. Your efforts during the process of revision are acknowledged and I hope you also are pleased with the final result. We appreciate being able to publish your work and look forward to seeing your paper online as soon as possible.

Kind regards,

José Moreira, MD, MSc

Academic Editor

PLOS ONE
---

## [Editor Report · Acceptance letter]

9 Oct 2020

PONE-D-20-14374R1 

Development of a multivariate prediction model of intensive care unit transfer or death: A French prospective cohort study of hospitalized COVID-19 patients 

Dear Dr. Maalouf:

I'm pleased to inform you that your manuscript has been deemed suitable for publication in PLOS ONE. Congratulations! Your manuscript is now with our production department. 

Kind regards, 

on behalf of

Dr. José Moreira 

Academic Editor

PLOS ONE